# Influences on drinking choices among Indigenous and non-Indigenous pregnant women in Australia: A qualitative study

**Sophie Gibson[1,2], Cate Nagle[3,4], Jean Paul[1,5], Leisa McCarthy[6], Evelyne Muggli** [1,2,4]*

**1** Reproductive Epidemiology, Murdoch Children's Research Institute, Parkville, Australia, **2** Victorian Infant Brain Studies, Murdoch Children's Research Institute, Parkville, Australia, **3** Centre for Nursing & Midwifery Research, James Cook University, Townsville, Australia, **4** Townsville Hospital & Health Service, Townsville, Australia, **5** Department of Paediatrics, The University of Melbourne, Parkville, Australia, **6** Menzies School of Health Research, Alice Springs, Australia

* evi.muggli@mcri.edu.au

**Data Availability Statement:** Data cannot be shared publicly because of confidentiality of participants. Some participants may be identified by their comments. Data are available from the

## Abstract

Despite women's awareness that drinking alcohol in pregnancy can lead to lifelong disabilities in a child, it appears that an awareness alone does not discourage some pregnant women from drinking. To explore influences on pregnant women's choices around alcohol use, we conducted interviews and group discussions with 14 Indigenous Australian and 14 non-Indigenous pregnant women attending antenatal care in a range of socioeconomic settings. Inductive content analysis identified five main influences on pregnant women's alcohol use: the level and detail of women's understanding of harm; women's information sources on alcohol use in pregnancy; how this information influenced their choices; how women conceptualised their pregnancy; and whether the social and cultural environment supported abstinence. Results provide insight into how Indigenous Australian and non-Indigenous pregnant women understand and conceptualise the harms from drinking alcohol when making drinking choices, including how their social and cultural environments impact their ability to abstain. Strategies for behaviour change need to: correct misinformation about supposed 'safe' timing, quantity and types of alcohol; develop a more accurate perception of Fetal Alcohol Spectrum Disorder; reframe messages about harm to messages about optimising the child's health and cognitive outcomes; and develop a holistic approach encompassing women's social and cultural context.

## Introduction

It is now well understood that heavy prenatal alcohol exposure is associated with a range of detrimental birth and developmental outcomes, such as Fetal Alcohol Spectrum Disorder (FASD). The severity of any of these outcomes depends on the level and gestational pattern of alcohol exposure and is also likely influenced by other maternal factors, such as nutrition, stress and genetics, as well as the postnatal psychosocial environment. Despite many longitudinal studies focussing on this topic, recent systematic reviews still find that the literature on

Murdoch Children's Research Institute Institutional Ethics Committee for researchers who meet the criteria for access to confidential data (contact Prof Jane Halliday at janehalliday.h@mcri.edu.au).

**Funding:** EM received funding from the Australian Institute of Health & Welfare for this work (https://www.aihw.gov.au); contract #0576_001. The funders had no role in study design, data collection and analysis, decision to publish, or preparation of the manuscript.

**Competing interests:** The authors have declared that no competing interests exist.

child outcomes with lower levels of exposure is inconclusive [1–3]. The ongoing debate of whether there is a 'safe' level of alcohol use in pregnancy has resulted in many official guidelines applying the precautionary principle and recommending complete abstinence in pregnancy.

The 2009 Australian National Guidelines to Reduce Health Risks from Drinking Alcohol recommend that it is safest for women to abstain from drinking alcohol completely throughout the duration of their pregnancy, while framing the message within the wider context of the available evidence. (Table 1) [4].

Data from two large national surveys of Australian women aged 18 to 45 years found that 34 to 49% of women consumed alcohol in pregnancy [5, 6], and that despite women's awareness that drinking alcohol in pregnancy can lead to lifelong disabilities in a child, nearly one third intended to drink alcohol in a future pregnancy [5]. Whilst knowledge of the potential harms of alcohol consumption during pregnancy is important, it is apparent that an awareness alone does not discourage some women from drinking when pregnant, perhaps stemming from their knowledge around the inconsistency of evidence around lower level alcohol use. The relatively high prevalence of alcohol consumption during pregnancy in Australia has also been reported in other high-income countries. An international cross-cohort comparison of the prevalence of alcohol use during pregnancy revealed that high rates of alcohol consumption in pregnancy, between 20 and 80%, were again evident despite knowledge of the guidelines recommending abstinence [7]. The levels of alcohol consumed in some sub-populations are of particular concern. For example, an Australian survey reported that of the 55% of Indigenous Australian women who consumed alcohol in pregnancy, nearly half drank at least two to three times per week and almost all consumed a minimum of seven standard drinks per occasion [8]. Similarly, a 2017 systematic review found that around one in five Indigenous women in North America drink at binge levels when pregnant [9].

While it is generally understood that frequent and heavy drinking among peers in populations with a low socioeconomic background is strongly associated with frequent and heavy alcohol use in pregnancy, [10] women who were highly educated, and/or with high incomes are also well represented among those who continue to drink in pregnancy, albeit at lower levels of consumption [5, 11–13]. Women with higher levels of education have more knowledge of the effects of alcohol use in pregnancy [14] and their drinking choices are likely to be based on an individualised risk perception rather than the blanket advice from guidelines and health professionals. It appears that a single health message to abstain from alcohol in pregnancy is not effective, especially in this population. Advice for pregnant women may need to be tailored to allow for social influences, attitudes, and personal experience, depending on the target population.

The objective of this qualitative study was to explore influences on pregnant women's decision making around alcohol use in a population with frequent and heavy peer drinking (i.e. in

**Table 1. Advice for women who are pregnant or planning a pregnancy (Australia)[1].**

| |
|---|
| • Not drinking alcohol is the safest option. |
| • The risk of harm to the fetus is highest when there is high, frequent, maternal alcohol use. |
| • The risk of harm to the fetus is likely to be low if a woman has consumed small amounts of alcohol before she knew she was pregnant or during pregnancy. |
| • The level of risk to the individual fetus is influenced by maternal and fetal characteristics and is hard to predict. |

[1] Australian guidelines to reduce health risks from drinking alcohol: Guideline 4 Pregnancy and breastfeeding. National Health and Medical Research Council, 2009.

two Indigenous Australian communities) and another of non-Indigenous pregnant women from a range of backgrounds. The aim was to better understand why messages to abstain may not always be effective with pregnant women and to inform a more tailored approach to health promotion.

## Methods

This study used data collected as part of a larger ongoing project, which aims to develop nationally consistent and comprehensive maternal and perinatal mortality and morbidity data collections in Australia, the National Maternity Data Development Project (NMDDP) [15]. The NMDDP included individual interviews and group discussions with Indigenous Australian and non-Indigenous pregnant women to explore their views on being asked about alcohol use as part of their routine maternity care and having this information reported at a national level. In this context, participants also reflected on their personal opinions and decisions on alcohol use in pregnancy and their understanding of the related harms. These conversations were analysed in the present study to address our objectives. The study is presented in line with the Consolidated Criteria for Reporting Qualitative Research (COREQ) [16] and was carried out in strict accordance with the Australian National Statement on Ethical Conduct in Human Research (2007). Ethical approval was granted by the Australian Institute of Health and Welfare (EO2015/3/196), the Human Research Ethics Committees of the Royal Children's Hospital (35173A&B), Cabrini Hospital (07-09-11-15), Mercy Health (R15-28), Western Health (35173A&B), Goulburn Valley Health (GVH32/15), the Research Sub-Committee of the Central Australian Aboriginal Congress Aboriginal Corporation, Alice Springs, and the Central Australian Human Research Ethics Committee (15–349).

Written consent was obtained from all study participants.

Women who were pregnant, aged 18 years or older and able to speak and write in English were invited to participate in the study. Recruitment was based on a convenience sample of women attending antenatal care at a date and time that researchers were in attendance. The sites were three public and one private health service in Victoria, Australia, and included socioeconomically disadvantaged and regional areas, as well as two Indigenous Australian settings; one remote service in the Northern Territory and one regional service in Victoria.

Women who met the selection criteria were approached by a member of the research team while they were waiting for their antenatal clinic appointment. Women who were interested in the study, provided written consent to participate and either took part in an individual interview held at a mutually convenient time (mostly immediately following their antenatal appointment) or in a group discussion later that day. Following consultation with clinic staff in the two Indigenous Australian communities, a personal choice to take part in an individual interview or group discussion was offered. This was to allow for freedom of expression where a young woman may not feel comfortable to express her own thoughts in the presence of an elder or may experience shame discussing the topic with others from their community.

The women taking part in this study had no prior relationship with any of the researchers. All individual interviews and group discussions began with brief introductions to the researchers' background, an explanation of the purpose of the study and an opportunity for the women to introduce themselves with their name and gestational age (e.g. *"Hi, my name is Anne and I am 18 weeks pregnant"*) and ask questions. This was followed by a guided discussion of the women's attitudes towards alcohol use during pregnancy, their understanding of the harms of drinking alcohol in pregnancy, and their views on collecting prenatal alcohol consumption information for clinical management and reporting purposes. Examples of guiding questions were: *What do you know about drinking alcohol in pregnancy? Where would you get your*

*information from about this sort of thing*?; *What is your experience of being asked by a maternity clinician about your alcohol drinking in pregnancy*? *What were your reactions*? *Remember, we don't want to ask you about your actual alcohol consumption, but to reflect on your own situation or that of friends.*; *How important do you think is it that maternity clinicians ask pregnant women about their alcohol use*? *Do you think that women in general would understand the reasons for being asked about this*?; and *How important do you think is it that information about alcohol use in pregnancy is collected and reported for all pregnant women across Australia*? The facilitator was free to ask follow-up questions to explore more deeply any topics raised in the discussion. The questions were pilot tested in a single discussion group with a convenience sample of five pregnant, or previously pregnant, women from the researchers host institution. Individual and group discussions were facilitated by a skilled qualitative researcher with expertise in the Indigenous Australian setting (JP). The presence of an experienced Indigenous Australian health researcher (LMcC) further ensured that the individual interviews in this setting were conducted respectfully and socially and culturally appropriate. In the groups, the facilitator conducted the discussion and ensured all participants were given the opportunity to contribute equally and a note taker was present to record contextual details and nonverbal expressions. All interviews and group discussions were audiotaped with participants' consent and transcribed verbatim with field notes added where relevant. Participant names and study sites were replaced with pseudonyms. Interviews and group discussions were not repeated and transcripts were not returned to participants. Aboriginal and/ or Torres Strait Islander people are respectively referred to in this paper as Indigenous Australian.

## Data analysis

Transcripts were analysed by a student researcher (SG) and her supervisor (EM) using inductive content analysis [17] with progressive feedback from the interview/discussion group facilitator (JP) on interpretation and coding. Content analysis is particularly useful to systematically identify specific messages in any type of social communication. It establishes the existence and frequency of concepts through inclusion or exclusion of content according to consistently applied criteria relevant to the research aims. Analysis involved repeated listening to recordings and reading of transcripts, coding and annotating the text, using the data management software NVivo 11 and on hard copies, with headings which represented manageable content categories. Data from individual interviews were not analysed separately as the responses did not identify different topics from those obtained in the groups. A process of selective reduction then produced an agreed analysis matrix, which consisted of hierarchical flow charts and diagrams to pictorially represent each heading and any possible connections between them. Data were then abstracted into this matrix in a dynamic process by further reviewing and refining headings with similar responses. These formulated categories became the final framework used to report results. Each category was named using a term that was 'content-characteristic' [17]. Using inductive content analysis enabled researchers to apply and discuss relevant theories to provide meaning and explain results which are included in the discussion section of this paper [18]. Participant feedback on the findings was not sought. A selection of representative quotes, with non-lexical utterances removed, is included in the results to illustrate the categories.

## Results

We interviewed 11 Indigenous Australian pregnant women and a further three took part in a group discussion. In the non-Indigenous setting, we conducted one individual interview and five small group discussions with two to three pregnant women in each group, totalling 14

**Table 2. Participating sites, number and gestation of participants.**

| Site | Code in quotes | Data collection method | Participant number | Interview time (mins) | Gestation (weeks) |
|---|---|---|---|---|---|
| Metropolitan public hospital 1 | MPUH1 | 1x individual interview | 1 | 27 | 29 |
| Metropolitan public hospital 2 (low SES) | MPUH2 | 2x group discussions | 5 | 30–33 | 34–39 |
| Rural/regional hospital | RH | 2x group discussions | 5 | 27–39 | 15–38 |
| Metro private hospital | MPRH | 1x group discussion | 3 | 27 | 20–31 |
| Remote Aboriginal community controlled health service[a] | RIHS1 | 3x individual interviews | 3 | 12–18 | 8–20+[b] |
| Rural Aboriginal community controlled health service | RIHS2 | 8x individual interviews; 1x group discussion | 11 | 15–28<br>34 | 8–35 |

[a] This health service was located in the Northern Territory. All other sites were located in the State of Victoria.

[b] Sometimes women were unsure about their gestational age. Researchers were advised not to specifically ask about this if the information was not volunteered.

participants (Table 2). Data collection was undertaken between November 2015 and March 2016. The time taken in individual interviews was between 12 and 18 minutes and group discussions took between 27 and 39 minutes with an average of 23 minutes taken across all.

The five main categories in our final analysis matrix were: (1) women's understanding of alcohol-related harm (understanding); (2) women's information sources on alcohol use in pregnancy (informing); (3) how this information influenced their choices (choosing); (4) how women conceptualised their pregnancy (conceptualising): and (5) whether their environment was supportive of abstinence from alcohol (enabling). Between three and five subcategories were identified to describe each categorical concept which increased understanding and generated knowledge about the topic. The conceptual framework for understanding the factors which influence drinking choices made by Indigenous Australian and non-Indigenous pregnant women is visualised in Fig 1.

## Women's understanding of harm (Understanding)

When asked about the harms of alcohol use in pregnancy, all women displayed an understanding and awareness that drinking alcohol was *"bad"*, and generally acknowledged that alcohol use could cause harm to their developing baby. Despite this knowledge, many participants were unclear about the nature of harm to the baby.

> *"I know that there is a number of things, I'm not exactly sure of what can happen but I know it can be dangerous."* (Participant 2 RIHS2)

> *"I know that there's some risks, but I don't know what they are. All I know is that I shouldn't be drinking."* (Participant 2 RH)

Some participants were able to describe one or more of the physical, social, emotional or behavioural symptoms such as wide eyes, slow learning and hyperactivity. Others named Fetal Alcohol Syndrome or 'FASD' (for Fetal Alcohol Spectrum Disorder) as the umbrella term for the effects from alcohol on the child, but then did not demonstrate an understanding of the condition's characteristics. Very few participants could both name and explain the disorder.

> *"From what I've heard, it's not good, there's conditions that can be involved with the development of the baby, and all. Like the actual, when the baby is born, there's serious brain function, cognitive development issues."* (Participant 1 MPUH1)

*"I guess I've heard a little bit about Fetal Alcohol Syndrome and all sorts of possible links back to things like ADHD, which really concerned me." (Participant 3 MPUH2)*

Many non-Indigenous women were aware that the research evidence for harm associated with low or occasional alcohol use was inconsistent and often described low level drinking as being safe.

*"There aren't really any good data on it, as far as in 'moderation' either, so. . ." (Participant 1 MPRH)*

*"He [the obstetrician] said that there's conflicting views out there and some say that one or two [drinks] is okay. He said; 'we just don't know enough about it'. He didn't say that you shouldn't [drink alcohol], he just put that information out there, which I would have known already, but it was nice to hear it from him." (Participant 2 MPRH)*

Participants often thought that harm was dependent on the timing of alcohol consumption, suggesting there was a *"dangerous period"* and a *"safe period"*. They generally agreed that it was important not to drink alcohol in the first 12 weeks of pregnancy and following this time, one or two occasional drinks would be unlikely to cause harm to their baby. When thinking about alcoholic drinks, most participants described a drink as being *"one glass of wine"* or *"a beer"*, showing only limited understanding of the concept of a 'standard drink'.

*"I do think that probably the general consensus is that people, a lot of people, wouldn't drink in the first twelve weeks at all and people might be more likely to after that." (Participant 2 MPRH)*

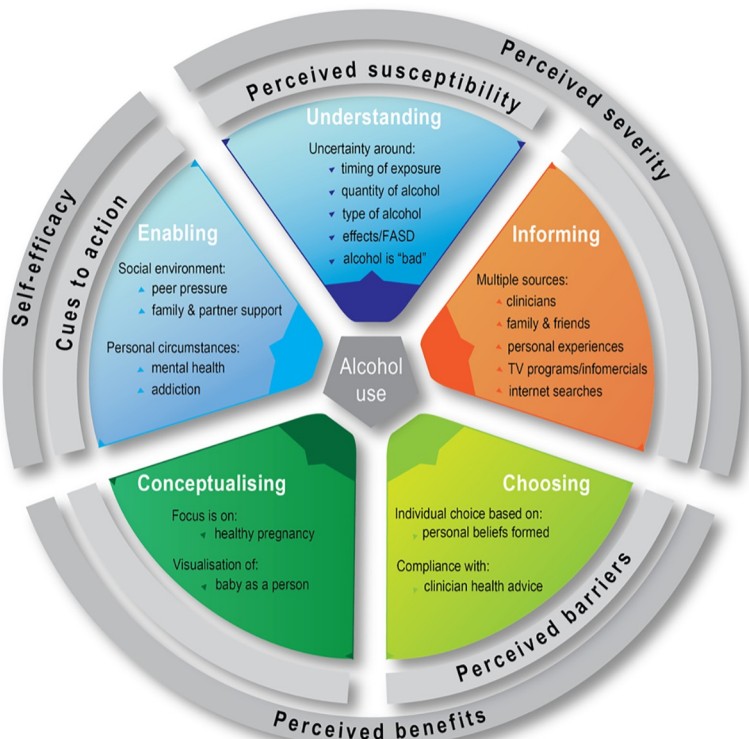

**Fig 1. Influences on pregnant women's drinking choices: Conceptual framework.**

*"I think people will be confused about the standard drinks. I don't think people are clear what a standard drink (is). They think that 'fill up a wine glass' is a standard drink." (Participant 2 MPRH)*

Some participants also believed that the type of alcoholic drink consumed played a role in the potential for harm, suggesting that drinks with lower alcohol content such as wine or beer, as opposed to spirits, were safer options.

*"My grandmother drank one glass of red wine each day with my uncles, and yet they turned out perfectly fine. But I think it's the hard stuff that you should avoid, how much percentage is in the actual alcohol or the wine, that's about it." (Participant 3 MPUH2)*

### Where women obtain information about alcohol in pregnancy (Informing)

All participants reported that their knowledge and understanding of harm from drinking alcohol in pregnancy stemmed from a variety of sources and not just from their midwife or doctor. Although clinicians were the primary source of information, the internet, television advertisements, and discussions with family and friends featured prominently.

*"Say I have access to a diverse group of mothers, whether it be from nationality, cultural, age group, and everybody has their own opinion on it." (Participant 2 MPUH2)*

*"Probably the internet. I'd probably look up research myself and I'm part of a couple of mums' groups online as well, so I'd probably to talk to them about it and see where they would think is a good place to get information as well." (Participant 1 MPUH2)*

*"Early on in the pregnancy, I didn't know I was pregnant for the first eight weeks so, I had a bit of alcohol, and I freaked out a little bit once I found out I was pregnant. I asked the GP and they were like 'no, should be fine' and I read a little, re-read over everything just to refresh, and then kind of came to a conclusion, nothing's going to be wrong." (Participant 1 MPUH1)*

*"I find a lot of stuff from America, and I will ask my mother in law, then my mama, because they both come from big families. If they don't know, I ask my doctor, or a chemist or someone around me. For the young mums especially, I think the internet's the first option for everyone because it's there, you don't have to leave the house." (Participant 4 RIHS2)*

*"I've seen a lot of TV ads as well about FASD." (Participant 1 RIHS1)*

Indigenous Australian participants in particular, reflected on discussions they had with their parents or grandparents about drinking during pregnancy, whereas non-Indigenous participants spoke about observing the social pattern of alcohol use in pregnancy of their family and friends whose children were subsequently unaffected.

*"Mum's been a midwife, she was doing her midwifery course when I was in her belly, so she's always been pretty open about everything and she tells me her views on things." (Participant 2 RIHS1)*

*"I was told through family, like what happens when you drink during pregnancy and stuff. They educated me before I went to a midwife sort of thing." (Participant 6 RIHS2)*

*"That's something that you do hear a lot; "Oh my mum said that she did this and we're all fine", that sort of thing. I guess it just depends what sort of people you've got around you,*

*whether you're hearing that constantly or whether you're hearing the other side of it." (Participant 1 MPUH2)*

*"I think mine's more the influence of other people, what they have done. I know people who have smoked and drank all through pregnancies and had healthy babies. But obviously there is always a risk involved."(Participant 2 MPUH2)*

Many Indigenous Australian participants mentioned that they had seen children affected by prenatal alcohol exposure within their community, family or workplace, and that this raised their awareness about the condition.

*"Others that have lived closer to big cities, they see it at school. You do see kids there that have been affected by Fetal Alcohol Syndrome. So, I think as well we're a bit exposed to it at school." (Participant 11 RIHS2)*

*"I guess my mum works with children with FASD, so I understand what happens when you drink during pregnancy. But I also think that there are people out there, that probably don't really understand the risks." (Participant 3 RIHS1)*

## How this information influenced their choices (Choosing)

Study participants used all information available to them to inform their decision-making. Indigenous Australian participants had generally seen evidence of the consequences of drinking in pregnancy and listened to their health practitioner's advice to abstain from drinking alcohol. However, most Indigenous Australian participants also voiced that some women may still believe alcohol use to be safe as they could not understand why anyone would drink if they knew there was a risk of harming their baby's health.

*"You know mothers are probably just thinking about; 'Oh well my mum did it with me, and her mum did it with her…" (Participant 3 RIHS2)*

*"Yeah, like they always say, "It never hurt your mum, or it never hurt you, we used to smoke all the way through our pregnancy so…'." (Participant 4 RIHS2)*

*"My mum, when I'm stressed, says 'it's okay if you have one'. I'm not comfortable doing that, my main reason is, because when I was pregnant with my twins, I was five months before I even found out, and I had been drinking and smoking that entire time. They came 10 weeks early and I carry an incredible amount of guilt because I didn't pay attention to myself and it feels like it's my fault that they came early." (Participant 2 RIHS2)*

In contrast, many non-Indigenous participants agreed that, while not drinking at all in pregnancy is safest, one or two occasional drinks would not be harmful. They were happy to take on board the information and advice given by their clinicians, but explained that their decision incorporated their own observations and 'research', the latter of which indicated a level of understanding of the conflicting evidence for harm with lower levels of alcohol use. All of this led to a thinking that whether to drink alcohol when pregnant was an individual choice.

*"I think there's conflicting information out there, but I will be of the opinion that none is safer, from what I have read." (Participant 2 MPRH)*

*"I think it's very individual. Some people, some of my girlfriends, have had the occasional drink at a wedding or something and I wasn't too concerned about it. But not drinking regularly or frequently or anything." (Participant 1 MPRH)*

*"They'll usually say the first twelve weeks is your dangerous period to be having alcohol. Then they're usually having it afterwards, but then some have had it right through, so it's their own choice really." (Participant 1 MPUH1)*

## How women conceptualise their pregnancy (Conceptualising)

When reflecting on their drinking choices, the women in our study spoke about their pregnancy in different ways, which also factored into their decision making. Women who spoke about their health and the health of their pregnancy, were more likely to also talk about making individual choices based on their own observations of the drinking behaviour of other pregnant women whose children developed normally despite having been exposed to some level of alcohol.

*"I just decided what I wanted for me, that was all." (Participant 2 MPUH2)*

*"I think people understand you're asked questions [about alcohol] to assess your health and the health of your pregnancy and I think people understand that.' (Participant 2 MPRH)*

In contrast, women who used language that was more directly connected to the developing fetus, such as the *"little baby inside"*, tended to emphasise that abstinence was very important. This language was used predominantly by Indigenous Australian women, but also by some women in rural or low socioeconomic settings.

*"Whatever you're eating and drinking, that's what your baby is drinking and eating as well . . . It's going inside your belly where the baby is." (Participant 5 RIHS2)*

*"Even though they've got an addiction and they need help, you know, who's helping that little baby inside?" (Participant 3 RIHS1)*

## Whether the woman's environment supports abstinence (Enabling)

Whilst Indigenous Australian participants acknowledged that some women in the community, particularly first-time mothers and those from remote communities, may not know about the harms of drinking in pregnancy, they believed that an inability to abstain in pregnancy related more directly to the influence of their social environment.

*"Whether it's a choice, or peer pressure, or they're following someone else." (Participant 8 RIHS2)*

*"If they want to just stop, I believe that they'd stop, but then again, it could be like other issues too. Just say that they have a partner, and they see them drink, then they feel like drinking because they don't want to be left out. There are all other things that come and play with it as well." (Participant 8 RIHS2)*

Several Indigenous Australian women explained that it was common for pregnant women in their community to have *"other stuff"* going on, such as mental health issues, addiction and domestic violence.

*"Especially for mums with domestic violence or if there's something with any type of trauma or issues, that may have happened to them, you know other stuff, and [they're] self-medicating. (Participant 3 RIHS2)*

*"Most of the time, if a woman is drinking during pregnancy, she's quite addicted to it. It's an addiction." (Participant 3 RIHS1)*

They also reflected on having the support of their family and/or partner and the protective value of strong culture. They felt that a lack of community, family and partner support was a clear risk factor for pregnant women to continue their drinking, and that not having a *"safe place"* to stay was also a risk factor.

*"If you have a safe place to go to, where there's no drinking, then I guess its ok. But it's hard for women that don't have a safe place to go, or somewhere to go where there's not people drinking. . .." (Participant 1 RIHS1)*

*"If you have supportive, strong family that are close and help each other, that's really good." (Participant 1 RIHS1)*

Indigenous Australian participants also thought that young pregnant women in particular were vulnerable to drinking because of a high frequency of unplanned and unwanted pregnancy and trying to keep up a social connection with their friends.

*"Especially if they're teenagers, all of their friends are teenagers and all of their friends are out drinking. They want to follow their friends and drink." (Participant 1 RIHS1)*

Although these points were predominantly raised by the Indigenous Australian women, some non-Indigenous participants also proposed social and environmental factors.

*"I guess someone who is drinking really heavily, perhaps they need some help. Maybe they need to see a counsellor or something like that, for an underlying issue, if they do have a problem with substance abuse." (Participant 1 MPUH2)*

*"Violence, which can lead to maybe the midwife referring on, or like depression or addiction. If something else is going on there as well, which could help with the treatment of, not only the mother, but the baby as well, and keeping the baby safe." (Participant 1 MPUH1)*

Further, the social importance of alcohol use, peer-pressure, and not being ready to disclose their pregnancy to others was thought to impact a pregnant woman's ability to abstain from alcohol.

*"We are the first in our group to try fall pregnant and it was something that obviously got; 'oh why aren't you drinking? Are you pregnant?'" (Participant 1 MPUH2)*

## Discussion

This study found some specific influences on pregnant women's alcohol use, which helped to explain why a message promoting abstinence is not always effective. This information may assist clinician's conversations about alcohol use in pregnancy and facilitate women to make healthy decisions. Women appeared to know that drinking alcohol when pregnant can be

detrimental to the developing baby, but when asked to describe the nature of harm in relation to alcohol use patterns or the effects on the child, women were usually uncertain. All women used the information available to them to make their decisions, but some placed great importance on individual choice; mostly because of their awareness of the limited evidence around harm from lower levels of alcohol use, but also perhaps rationalising their preference to drink alcohol while pregnant. The decisions to abstain in the other group of women, which predominantly comprised the Indigenous Australian participants, were based on their understanding of the responsibility of having a growing baby inside them, listening to the health care provider, and being exposed to the harmful consequences of drinking alcohol in pregnancy. Additional factors which enabled abstinence in both groups, included the women's social environment and the support of a family or in the Indigenous Australian setting, a strong culture.

The Health Belief Model [19] is a useful framework to assist in the understanding of the relationship between health beliefs and health practices and provides a good fit to explain our study findings and propose potential strategies for change below (see Fig 1). The model addresses: a) a person's perception of a threat posed by a health problem, such as their susceptibility or the severity of the condition (i.e. harm from alcohol use in pregnancy); b) the benefits of, or barriers to, avoiding the threat by taking a recommended action (i.e. abstaining from alcohol in pregnancy); and c) the factors that prompt the recommended health action and a person's ability to take such action (i.e. abstinence within a social and/or cultural context) [20].

## Perceived susceptibility to harm

Firstly, the idea that some alcohol was safe to drink after the first trimester, or that spirits were more harmful than wine, influenced women's individual views of their susceptibility to harm. Misconceptions about the safety or supposed health benefits of different types of alcoholic drink are common, whether in the context of pregnancy, [21, 22] or in the general population [23]. Further, it is well established that there are many misconceptions about the 'standard alcoholic drink'. The 'standard drink' or 'unit of alcohol' is a concept originally developed by the UK Government for their 1987 guidelines on safe drinking and adopted by the World Health Organisation and many countries around the world [24]. Despite standard drink labelling on alcoholic drink beverages, the women in our study equated one glass of wine or 'a beer' with one single drink, even though for example, a 375millilitres can of full-strength beer contains 1.5 standard drinks of alcohol in Australia.

## Perceived severity of harm and benefits from abstaining

Secondly, some women perceived the severity of harm from occasional alcohol use to be low. This was usually based on their personal observations of the behaviour of family and friends, and an understanding of the lack of convincing research evidence on harm from low consumption patterns. Consequently, beliefs about the benefits of abstaining from alcohol completely were also low in this group of (non-Indigenous) women and the barriers to taking such action, for example when at a social event, were seen to outweigh any risks. Together, these perceptions permitted nuanced decisions by individual women about the quantity of alcohol was without risk of harm, even if they received best practice health messages advising abstinence. In contrast to the group of women who were making individual decisions about how much was safe to drink, pregnant women who commonly saw heavy alcohol use in their community were more likely to believe that there could be serious harm from alcohol to their child and that complete abstinence was important.

## Perceived barriers to abstaining

Thirdly, despite an understanding of the potential harm from alcohol consumption and the clinician's advice, a pregnant woman's social environment may limit her ability to abstain. Previous research with Indigenous Australian communities indicates that while Indigenous Australian women are less likely to drink alcohol in pregnancy than non-Indigenous women, if they do drink, it is done so at risky levels [6, 8]. This was supported by our conversations with Indigenous Australian women who spoke about the need for a strong, supportive family in environments such as public housing town camps, where high-risk alcohol use is common.

All women in our study perceived that there was an expectation on them to drink alcohol in social situations when not pregnant and explained that this influenced their decision or ability to abstain during pregnancy. The pressure to comply with such social norms, especially in early gestation and before the pregnancy is disclosed to others, are well documented in Australian and other Western countries with a similar alcohol use culture [25, 26].

## Cues to action and self-efficacy

Lastly, the outcomes from discussions with pregnant women in our study illustrate that abstinence from alcohol has many facets, undermining a blanket public health message in many instances. To improve women's knowledge of the harms from alcohol and their own susceptibility, health advice should include specific education to correct misinformation about 'safe' gestational timing, and increase understanding of the standard alcoholic drink concept to address the mistaken belief that some types of drink are less harmful or even beneficial to one's health. Further, we need to help develop a more accurate perception of FASD and provide a clear message that is evidence-based. Many women are aware of the current lack of evidence for harm associated with low or occasional alcohol use and infer from this that it is safe to drink some alcohol. While risks to the fetus may be low with lower levels of alcohol consumption, it is not currently known if there is a safe threshold of exposure. Alcohol is an established teratogen, acting either directly, or through its metabolites, and affecting the regulation of cellular functions [27, 28], and while exposure may not necessarily result in adverse clinical outcomes for the child, this does mean that fetal development is influenced at a biological level in response to alcohol. Women may think about avoiding potential adverse cognitive outcomes in their child when making decisions about how much to drink when pregnant, but prenatal alcohol exposure may also result in increased vulnerabilities to less clearly defined later mental health problems, such as depression and anxiety [29]. Further, our own research showed an association between any prenatal alcohol exposure and facial shape of one year-old children, resulting in imperceptible, yet measurable, flattening of the midface and upturning of the nose [30], adding weight to the growing body of evidence on the influence of alcohol on all stages of fetal development [31, 32]. In light of this, we may need to reframe discussions around harm prevention or whether there is a potentially 'safe' threshold, to messages about the importance of alcohol abstinence in optimising health and cognitive outcomes for the unborn child. At the population level, FASD-specific mass media campaigns, based on proven behaviour change principles and with messages which combine threat (addressing perceived susceptibility and severity) and self-efficacy (promoting confidence in ability to abstain) have been shown to be effective in the past [33, 34].

For women with unsafe alcohol use or whose social and cultural environment makes abstinence difficult, clinicians can play an important role in supporting and encouraging reduction in intake. There is good evidence that brief interventions can be effective. These usually follow the '3 As' of 'Assess, Advise and Assist' and include building rapport, verbal reinforcement, goal setting to build confidence, and assisting with personal circumstances [35, 36]. Building

rapport and providing culturally safe and holistic antenatal care is especially important for Indigenous Australian women who may experience a disproportionate number of adverse circumstances [37, 38].

## Methodological considerations

This was a qualitative research study comprising a convenience sample of pregnant women at low risk of complications, attending a variety of antenatal care settings and this need to be considered in the transferability of our findings. We offered group discussions to encourage participants to analyse their opinions more deeply within the group dynamic and, on the advice of Indigenous Australian clinic staff, individual interviews where women may not have felt comfortable to speak freely in the presence of their peers. The findings presented arose from data generated in both group and individual discussions with pregnant women, which may have affected the nature of conversations. However, the combination of both types of data collection allowed for topics to arise in a group interaction as well as the voicing of opinions in private, neither of which evoked distinguishable content. While findings may be specific to the 28 women taking part and possibly their broader interactions, our data analysis confirmed that data saturation was reached, and no new topics arose that warranted further investigation.

The influence of unintended pregnancy, or the time period before pregnancy awareness in general on alcohol use was not considered specifically in this study and may require additional approaches, such as FASD-specific public health initiatives.

## Acknowledgments

We acknowledge the support of Central Australian Aboriginal Congress Aboriginal Corporation, Mallee District Aboriginal Services, Cabrini Health, Western Health, Mercy Health and Goulburn Valley Health and wish to thank all clinic managers and other staff for their generous help with recruitment, organising rooms, and allowing us to interview their clients. We also thank Clare Morrison and Taryn Charles for their assistance with the interviews and discussion groups and Rigan Tytherleigh, Rachel Gordon and Joanne Kennedy for their transcriptions, Professor Della Forster for her critical input into drafting of the manuscript and Romano Studer for his assistance with the figure design. Most of all, we would like to thank all study participants for their time and effort in assisting us with this research.

## Author Contributions

**Conceptualization:** Cate Nagle, Evelyne Muggli.

**Data curation:** Sophie Gibson, Jean Paul, Evelyne Muggli.

**Formal analysis:** Sophie Gibson, Evelyne Muggli.

**Funding acquisition:** Evelyne Muggli.

**Investigation:** Sophie Gibson, Jean Paul, Evelyne Muggli.

**Methodology:** Cate Nagle, Jean Paul, Leisa McCarthy, Evelyne Muggli.

**Project administration:** Jean Paul, Leisa McCarthy, Evelyne Muggli.

**Resources:** Evelyne Muggli.

**Supervision:** Evelyne Muggli.

**Validation:** Jean Paul, Evelyne Muggli.

**Visualization:** Sophie Gibson, Evelyne Muggli.

**Writing – original draft:** Sophie Gibson.

**Writing – review & editing:** Sophie Gibson, Cate Nagle, Jean Paul, Leisa McCarthy, Evelyne Muggli.

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
