## [Decision Letter · Decision Letter 0]

9 Jan 2020

PONE-D-19-29063

Pregnant women’s understanding and conceptualisations of the harms from drinking alcohol: a qualitative study

PLOS ONE

Dear Ms Muggli,

Thank you for submitting your manuscript to PLOS ONE. After careful consideration, we feel that it has merit but does not fully meet PLOS ONE’s publication criteria as it currently stands. Therefore, we invite you to submit a revised version of the manuscript that addresses the points raised during the review process.

Please attend to both reviewers comment but in particular the issues raised by reviewer 1.

We would appreciate receiving your revised manuscript by Feb 23 2020 11:59PM. To enhance the reproducibility of your results, we recommend that if applicable you deposit your laboratory protocols in protocols.io, where a protocol can be assigned its own identifier (DOI) such that it can be cited independently in the future. For instructions see: http://journals.plos.org/plosone/s/submission-guidelines#loc-laboratory-protocols

We look forward to receiving your revised manuscript.

Kind regards,

Catherine Haighton, PhD

Academic Editor

PLOS ONE

Journal Requirements:

Reviewers' comments:

Reviewer's Responses to Questions

**Comments to the Author**

1. Is the manuscript technically sound, and do the data support the conclusions?

Reviewer #1: Yes

Reviewer #2: Yes

2. Has the statistical analysis been performed appropriately and rigorously? 

Reviewer #1: N/A

Reviewer #2: Yes

3. Have the authors made all data underlying the findings in their manuscript fully available?

Reviewer #1: No

Reviewer #2: Yes

4. Is the manuscript presented in an intelligible fashion and written in standard English?

Reviewer #1: Yes

Reviewer #2: Yes

5. Review Comments to the Author

Reviewer #1: The paper provides an interesting account of pregnant women's understanding and conceptualisation of the harms of drinking alcohol. This is an important topic. Whilst the paper has many merits, further work is required to strengthen. My main concern is the assumption throughout the paper that any alcohol is problematic. This is not accepted universally within the field, and there is a lack of convincing evidence for harm below 5 units of alcohol. Greater specificity is required in the intro when discussing any alcohol (including very low levels) and when the literature referenced is concerned with heavier levels. This issue also requires greater engagement in the discussion. In my view, it is not sufficient for the authors to cite one study (their own) to show that the women are 'wrong' when perceiving small amounts of alcohol to be low risk, given a large volume of research which has found little evidence for risk associated with lower levels (see systematic reviews: Henderson, J., R. Gray, and P. Brocklehurst, Systematic review of effects of lowmoderate prenatal alcohol exposure on pregnancy outcome and Gray, R., Low-to-moderate alcohol consumption during pregnancy and child development – moving beyond observational studies). Given the stated objectives of paper to examine influences, I was disappointed that the authors did not engage with the discrepancy that exists between a blanket no alcohol message and much of the evidence that the women are able, and choose to access. This discrepancy undermines the public health message and could have been drawn out in the findings (informing) and in the discussion. The authors could then also explore how best to inform in light of this discrepancy in evidence/advice. Presumably if the authors believe the women need to have their misinformation corrected, proving evidence to the women of the harm is the way to go. This would be most unhelpful and may result in adverse outcomes (unplanned pregnancies where alcohol was consumed prior to knowledge of pregnancy being unduly concerned/choosing to terminate for fear of impact caused, greater mistrust of information, decreased disclosure, stigma and judgement). Whether the authors accept that any alcohol is low risk during pregnancy or not, the women appear to be influenced by the community/informal beliefs and the lack of evidence showing harm at low levels of consumption.

There appears to be a mismatch between the title and the objectives. The title is focused upon the women's understanding whilst the objectives is concerned with behaviour influences. The objectives would seem much broader than the title implies.

Methods - further details are needed about the 'group discussions'. The language used to describe this method is not consistent and at times is referred to as a group discussion (which implies focus groups) and at others is referred to as a group interview. Why was this approach used? Some justification is given for times when it was NOT used (e.g. to increase comfort) but none for the benefits of interviewing three women together. How were the groups formed? Was there any thought about characteristics that would make the groups appropriate/helpful to promote discussion, did the women know each other, was there group interaction (as focus group)? I understand the authors used convenience sampling. A number of factors are likely to be relevant to the women's influences. Is there any information about sample variation beyond Indigenous Australian/non Indigenous and weeks gestation? Women who choose to drink a little during pregnancy, those who drink heavily and those who abstain are likely to understand and conceptualise harm from alcohol very differently. It may have been more helpful to take a purposive approach to sampling. However, as this approach was not taken, can the authors comment of levels of alcohol use within the sample? Or provide justification as to why this was not discussed.

The findings are quite thin and some of the interviews short (10 mins). This suggests a lack of depth in the data. What was the mean interview length? If this is closer to 30 mins this would provide some reassurance.

I think the table of quotes which is presented disconnected to the analysis is unhelpful and it is unclear how the 'representative sample' has been selected. I would encourage the authors to integrate these within the findings. I am unclear what is the table and what is the figure.

The discussion would benefit from greater depth and engagement with the literature. In particular, as discussed above there is opportunity to engage with the mismatch between what the current approach is (aimed at influencing the women's drinking) and how the women understand harm. Multiple subheadings within the discussion interrupts the discussion.

Reviewer #2: This appears to be a very interesting and potentially useful study. The importance of the topic cannot be overstated. The use of the HBM framework to design and analyze the study is especially compelling. The presentation of the data and the Discussion sections are very thought-provoking and offer useful clinical information.

However, I did have some questions/comments:

1) The title is very general, but the sample is a very specific group of women. A title that provided a better characterization of population from which the sample is drawn might make the paper more applicable and address issues of diversity.

2) Along the same lines, data on how many women were available and how many actually agreed to participate would provide a better basis for seeing this very small sample as part of a larger context.

3) The Introduction could benefit from two changes. First, there is literature on why people in general make poor drinking decisions even when they have full awareness of the potential negative consequences. Some review of this literature would be helpful. Second, the interviews appear to be informed by the Health Belief Model, but HBM is not really discussed in the introduction. I would encourage a little more of a theoretical basis for paper to appear in the Introduction, rather than waiting until the Discussion.

6. PLOS authors have the option to publish the peer review history of their article (what does this mean?). If published, this will include your full peer review and any attached files.

Reviewer #1: No

Reviewer #2: No

---

## [Author Response · Author response to Decision Letter 0]

3 Mar 2020

We have addressed the reviewers' feedback in the attached letter (Response to Reviewers).

---

## [Decision Letter · Decision Letter 1]

14 Apr 2020

Influences on drinking choices among Indigenous and non-Indigenous pregnant women in Australia: a qualitative study

PONE-D-19-29063R1

Dear Dr. Muggli,

We are pleased to inform you that your manuscript has been judged scientifically suitable for publication and will be formally accepted for publication once it complies with all outstanding technical requirements.

With kind regards,

Catherine Haighton, PhD

Academic Editor

PLOS ONE

Additional Editor Comments (optional):

Reviewers' comments:

Reviewer's Responses to Questions

**Comments to the Author**

1. If the authors have adequately addressed your comments raised in a previous round of review and you feel that this manuscript is now acceptable for publication, you may indicate that here to bypass the “Comments to the Author” section, enter your conflict of interest statement in the “Confidential to Editor” section, and submit your "Accept" recommendation.

Reviewer #2: All comments have been addressed

2. Is the manuscript technically sound, and do the data support the conclusions?

Reviewer #2: Yes

3. Has the statistical analysis been performed appropriately and rigorously? 

Reviewer #2: Yes

4. Have the authors made all data underlying the findings in their manuscript fully available?

Reviewer #2: Yes

5. Is the manuscript presented in an intelligible fashion and written in standard English?

Reviewer #2: Yes

6. Review Comments to the Author

Reviewer #2: I appreciate that the authors reviewed my comments carefully and gave due consideration to their answers. I think they did an excellent job in responding and I am satisfied.

7. PLOS authors have the option to publish the peer review history of their article (what does this mean?). If published, this will include your full peer review and any attached files.

Reviewer #2: No

---

## [Editor Report · Acceptance letter]

20 Apr 2020

PONE-D-19-29063R1 

Influences on drinking choices among Indigenous and non-Indigenous pregnant women in Australia: a qualitative study 

Dear Dr. Muggli:

I am pleased to inform you that your manuscript has been deemed suitable for publication in PLOS ONE. Congratulations! Your manuscript is now with our production department. 

With kind regards,

on behalf of

Dr. Catherine Haighton 

Academic Editor

PLOS ONE